# Association of Exposure to a Combination of Ergonomic Risk Factors with Musculoskeletal Symptoms in Korean Workers

**DOI:** 10.3390/ijerph17249456

**Published:** 2020-12-17

**Authors:** Jungsun Park, Yangho Kim

**Affiliations:** 1Department of Occupational Health, Catholic University of Daegu, Gyeongsan 38430, Korea; jsunpark@chol.com; 2Department of Occupational and Environmental Medicine, Ulsan University Hospital, University of Ulsan College of Medicine, Ulsan 44033, Korea

**Keywords:** musculoskeletal abnormalities, disorder, ergonomics, risk factors, exposure

## Abstract

This study examined the relationship of musculoskeletal symptoms with exposure to a combination of ergonomic risk factors at work and the possible ameliorating effect of enough time to rest during working hours or between consecutive shifts in Korean workers. Data were from the 2017 Korean Working Conditions Survey. Workers exposed to ergonomic risk factors were more likely to report musculoskeletal symptoms than those without exposure, and exposure to more ergonomic risk factors increased the probability of musculoskeletal symptoms. Workers who had the opportunity to rest when desired and those who had enough time to rest between consecutive shifts were less likely to report musculoskeletal symptoms. In conclusion, workers exposed to more ergonomic risk factors had an increased risk for musculoskeletal symptoms, and providing enough time to rest and recovery to workers reduced the risk of musculoskeletal symptoms.

## 1. Introduction

The European Risk Observatory Report considers musculoskeletal disorders (MSDs) to be the most common work-related disorder in Europe [1]. In Korea, MSDs are also the most common work-related disorder, and the number of reported MSDs has also increased over time [2]. In addition to their effects on workers, work-related MSDs (WRMSDs) are also a financial burden to businesses and society. In an effort to reduce the prevalence of WRMSDs, Korean regulations from 2003 state that an employer at a workplace with even a single ergonomic hazard (musculoskeletal burden) among 11 specified hazards should assess the risk of each hazard once every 3 years and provide notification to the Ministry of Employment and Labor [2].

MSDs are more likely to occur when a muscle is fatigued due to repeated movements and when there is insufficient time for recovery. Armstrong et al. [3] presented a conceptual model to explain the complex relationship of WRMSDs with multiple workplace risk factors and individual factors. Their model suggested that repetitive or sustained microtrauma (mechanical or physiological) reduced the integrity and function of specific tissues and structures within the musculoskeletal system. Their model also characterized this dynamic relationship as a series of cascading events, in which the response may be viewed as another “dose” that improved or decreased individual muscular capacity. The amount, duration, and frequency of loads imposed on muscles, as well as the recovery time, determine whether tolerance increases due to the effect of training or decreases and leads to an MSDs [3]. Muscular damage can result from high tension development, especially in eccentric contractions which can lead to rupture of the muscle fiber Z-line [4]. Such muscle work can also cause a large but delayed increase in serum creatine kinase [5], and this increase is related to excessive occupational work [6,7]. Such changes are common findings in individuals who report muscle soreness, and are reversible if the muscle is provided with sufficient rest [3].

Many workers are exposed to diverse ergonomic risk factors, especially those employed in construction, agriculture, manufacturing, wholesale and retail trade, and human health and social work [8]. Moreover, many workers are simultaneously exposed to several ergonomic risk factors, so it is difficult to attribute a WRMSD to a single workplace risk factor [1]. Thus, a more general assessment of the ”strenuousness at work” could help develop new policies that consider the multifactorial causality of WRMSDs [1]. Choi and Lee found that simultaneous exposure to “repetitive work”, “awkward posture”, and “heavy load” explained 38% of WRMSDs in the “transportation machinery and equipment manufacturing” sector [9].

However, very few epidemiological studies have investigated the relationship between exposure to a combination of ergonomic risk factors and WRMSDs, and some of the results have been contradictory. The interaction from the simultaneous exposure to multiple ergonomic risk factors can also increase the risk of WRMSDs [10], although the nature of these interactions require clarification [11]. However, an epidemiological study suggested that the combination of ergonomic risk factors did not affect back pain, but did affect elbow and hand/wrist pain [12]. A recent study in Korea reported that the combined effect of ergonomic risk factors was less likely to be associated with work-related lower back pain [13], but another study showed that simultaneous exposure to inadequate posture and whole-body vibration increased the risk of back pain [14].

We, therefore, examined musculoskeletal symptoms in Korean waged workers. We first determined the effect of exposure to combined ergonomic factors at the workplace on musculoskeletal symptoms after adjustment for covariates, and then determined the possible ameliorating effect of enough time to rest during working hours or between consecutive shifts on musculoskeletal symptoms.

## 2. Materials and Methods

### 2.1. Design and Data Collection

All data were from a secondary analysis of the fifth Korean Working Conditions Survey (KWCS), which was collected from June to September 2017 by the Korea Occupational Safety and Health Agency (KOSHA) [15]. This triennial survey assesses workplace conditions and exposures to hazards and health problems that are related to work. The study population was representative of Korean workers (i.e., individuals who worked for pay or profit for 1 h or more in the week prior to the interview), and all individuals were 15 years old or more. Retirees, the unemployed, homemakers, and students were excluded. This study had a multistage and stratified design of randomly selected workers from the same enumeration districts used for the 2010 Korean population and housing census. 

The survey consisted of 50,205 in-person household interviews. The sample was representative of the economically active population of Korea due to statistical weighting. Thus, the study participants and economically active population had similar region, locality, and population of residence; sex; age; employment status; and industry of employment. A prior study indicated the KWCS had high content validity and reliability [16].

A total of 50,205 subjects were initially interviewed. After elimination of foreign workers, employers, and the self-employed, there were 30,060 study subjects. After the study was explained, each subject provided written informed consent prior to participation. Institution and Ethics approval and informed consent: All participants provided written informed consent. This research was approved by the Institutional Review Board (IRB) of Ulsan University Hospital (IRB File No. 2020-06-007).

### 2.2. Measurements

The survey evaluated health status, self-reported exposure to vibration, ergonomic risk factors, and other working conditions.

#### 2.2.1. Dependent Variable

The following question was used to assess musculoskeletal symptoms: “Have you suffered from back pain, upper limb pain, or lower limb pain during the past 12 months?” A response of “yes” indicated the presence of a musculoskeletal symptoms. Each musculoskeletal symptom was separately recorded as “back pain”, “upper extremity pain”, or “lower extremity pain”.

#### 2.2.2. Demographic, Socioeconomic, and Workplace Factors (Independent Variables) 

The following demographic and other workplace-related factors were recorded: sex, age in years, education level (no high school graduation, high school graduation, at least some college), monthly income in USD (<1000, 1000 to less than 2000, 2000 to less than 3000, 3000 to less than 4000, and ≥4000), and average weekly working hours. We divided the subject’s age into young (<40), middle (40 to 49), aging (50 to 59), and aged (≥60) considering comparability and age distribution. Average weekly working hours were classified as short working hours (<40 h), moderate working hours (40–47 h), long working hours (48–59 h), or excessively long working hours (≥60 h). Forty hours per week is the standard working hour, and weekly working hours exceeding 48 are defined as long working hours [17]. Excessively long working hours are defined as ≥60 h in Japan and Korea considering the risk of Karoshi [18,19,20]. 

Each participant was classified into one of four occupational classes: “managers, professionals, and clerks”, “service and sales workers”, “skilled manual workers”, or “unskilled manual workers”. Labor was initially categorized as non-manual or manual, and manual labor was then categorized as skilled or unskilled. Participants who were skilled manual workers were in the following 3 major groups from the Korean Standard Classification of Occupations (KSCO) [21] and the International Standard Classification of Occupations (ISCO) [22]: “Skilled workers Related to Agriculture, Forestry and Fisheries”, “Craft and Related Trade Workers”, and “Workers Related to Equipment, Machine Operating and Assembling”. Participants who were unskilled manual workers had “Elementary Occupations” as described in the KSCO [21] and the ISCO [22]. Participants who were non-manual workers were “managers, professionals, and clerks” or “service and sales workers”. Because “service and sales workers” perform more physical labor for longer periods of time, they differ from “managers, professionals, and clerks”, who typically performed sedentary work in offices.”

#### 2.2.3. Ergonomic Risk Factors at the Workplace (Independent Variables)

Exposures to four different ergonomic risk factors were determined by asking: “Please tell me, using the scale (all of the time/almost all of the time/about three-quarters of the time/about half of the time/about one-quarter of the time/almost never/never), are you exposed at work to: (i) vibration due to use of hand tools or machinery; (ii) painful or tiring postures; (iii) lifting heavy loads; and (iv) repetitive hand or arm movements?” Each risk factor was classified as “non-exposure” (one-quarter or less of working hours) or “exposure” (half or more of working hours). There were two questions about resting. First, the ability to rest at the workplace when desired was determined by asking: “Can you take a break when you wish?” The possible answers were “always/most of the time” (classified as “yes”) or “sometimes/rarely/never” (classified as “no”). Second, enough time to rest between consecutive shifts was determined by asking: “In the last month, has it happened at least once that you had less than 11 h between the end of one working day and the start of the next working day?” The possible answers were “yes” or “no”. Concerning the organization of working time, Directive 2003/88/EC states that Member States shall take the measures necessary to ensure that every worker is entitled to a minimum daily rest period of 11 consecutive hours per 24 h period [23].

### 2.3. Statistical Analysis

SPSS version 20 was used for statistical analyses, and a *p*-value below 0.05 was considered significant. The chi-square test was initially used to compare variables of workers who did and did not report musculoskeletal symptoms (Table 1). Then, multiple logistic regression analysis was used to calculate adjusted odds ratios (aORs) and 95% confidence intervals (95% CIs) for the relationship of musculoskeletal symptoms with a combination of ergonomic risk factors, following adjustment for confounding by demographic and socioeconomic factors (age, gender, education, and income) and work-related factors (occupational class, weekly working hours, ability to rest when desired, and having time less than 11 h between consecutive shifts). In this analysis, an ordinal measure (from 0 to 4) was used to summarize exposure to the following ergonomic risk factors: vibration, awkward postures, lifting heavy loads, and repetitive movements. Workers exposed to 1 or more ergonomic risk factors were compared to workers without exposure (reference group).

## 3. Results

### 3.1. Socio-Demographic Characteristics of the Study Subjects

Table 1 shows the socio-demographic characteristics of the study subjects. A total of 48.2% of the study subjects were men, and 37.8% of them were at least 50 years old. Most study subjects had graduated from high school (34.9%) or college (52.5%). A total of 40.7% of study subjects had monthly incomes less than USD 2000. Among the study subjects, 40.1% were “managers, professionals, and clerks”, followed by “service and sales workers” (26.3%), “skilled manual workers” (17.9%), and “unskilled manual workers” (15.3%). A total of 53.2% worked moderate working hours, followed by long working hours (21.0%), short working hours (15.4%), and excessively long working hours (10.1%) (Table 1).

### 3.2. Factors Associated with Musculoskeletal Symptoms

Musculoskeletal symptoms in the back, upper extremity, and lower extremity were common in the following groups: female workers, aged workers, low educated and low income workers, manual workers, workers exposed to ergonomic risk factors, and workers with the shortest or longest weekly working hours, no ability to rest when desired, and/or time less than 11 h between consecutive shifts (Table 2).

### 3.3. Association of Workplace Factors with Musculoskeletal Symptoms

We performed a multivariate logistic regression analysis to identify ergonomic risk factors at the workplace that were associated with musculoskeletal symptoms in three anatomical regions, with adjustment for demographic, socioeconomic, and work-related factors as covariates (Table 3). The results indicated that workers exposed to ergonomic risk factors were more likely to report a musculoskeletal symptom in each anatomical region than those with no exposure (reference group). In addition, the aOR for a musculoskeletal symptom in each anatomical region increased as the number of ergonomic hazards increased from 1 (aOR = 1.497 to 1.667) to 4 (aOR = 3.893 to 4.938). These graded relationships were prominent in upper extremity pains.

We also analyzed other factors associated with musculoskeletal symptoms in each anatomical region. Women (aOR = 1.563 to 1.685) and older workers (aOR = 1.509 to 2.243) were more likely to report all musculoskeletal symptoms. Workers with the ability to rest when desired (aOR = 0.624 to 0.667) and those without time less than 11 h between consecutive shifts (aOR = 0.495 to 0.642) were less likely to report all musculoskeletal symptoms. In particular, those without time less than 11 h between consecutive shifts were 50.5% less likely to report back pain. Those who had more education (aOR = 0.575 to 0.430) were also less likely to report all musculoskeletal symptoms. Those who had higher monthly incomes (aOR = 0.747 to 0.682) were less likely to report back pain, but monthly income was unrelated to upper extremity pain and lower extremity pain. Skilled and unskilled manual workers (aOR = 1.312 to 1.610) were more likely to report all musculoskeletal symptoms. Service and sales workers (aOR = 1.369) were more likely to report lower extremity pain. Those who worked 60 h or more per week (aOR = 1.216 to 1.357) were more likely to report all musculoskeletal symptoms; those who worked 48 to 60 h per week were more likely to report upper extremity pain (aOR = 1.143); and those who worked 40 to 48 h per week (aOR = 0.838) were less likely to report lower extremity pain.

## 4. Discussion

The present study showed that many workers reported exposure to a combination of ergonomic risk factors that could lead to musculoskeletal symptoms. In particular, exposure to a greater number of ergonomic risk factors increased the odds ratio of musculoskeletal symptoms, upper extremity pain, in particular. However, few previous epidemiological studies have investigated the relationship between exposure to a combination of ergonomic risk factors and WRMSDs, and some of the results have been contradictory [10,11,12,13,14]. Our findings confirmed that WRMSDs are more likely to be caused by complex ergonomic hazards. We also found that workers who had enough time to rest during working hours or between consecutive shifts were less likely to report all the musculoskeletal symptoms. However, only a few papers reported the relationship between rest break and musculoskeletal pain. Armstrong et al. [3] reported that an MSD was more likely when a muscle was repeatedly fatigued and not given sufficient time for recovery. Wang et al. [24] found a strong association between the ratio of work time to recovery time within a day and neck-shoulder disorders. No previous paper reported the association between enough time to rest between consecutive shifts and musculoskeletal problems. The present study is the first to evaluate the relationship of musculoskeletal problems with enough time to rest during working hours and between consecutive shifts to the best of our knowledge. Many previous studies found that workers with long working hours had a higher incidence of musculoskeletal symptoms [25,26,27,28]. However, longer working hours do not necessarily lead to shorter resting time although excessively long work schedules may limit rest and recovery time. The present findings support this issue of long working hours. Taken together, primary prevention of WRMSDs should consider reducing the exposure of workers to a combination of ergonomic risk factors and providing workers with sufficient time for muscle recovery within shifts and/or between consecutive shifts.

The present study found that women were more likely to report musculoskeletal symptoms than men. Previous studies also showed that employed women were 2 to 5 times more likely to report musculoskeletal pain than men [29,30,31]. This gender difference in reporting musculoskeletal symptoms may be because it is more socially acceptable for women than men to report pain or because women have lower thresholds for pain than men [32,33]. We also found that musculoskeletal symptoms in workers were associated with older age. This is likely because elderly workers are more likely to have reduced muscle strength and elasticity and a more limited range of motion in their joints, and are, therefore, more likely to complain of musculoskeletal symptoms than young workers [34,35,36].

We assessed the association of occupational categories with musculoskeletal symptoms, by calculating the aORs for musculoskeletal symptoms for each occupational category relative to a reference group of “managers, professionals, and clerks”, who mostly performed sedentary work in offices. Manual workers were more likely to report all the musculoskeletal symptoms. However, service and sales workers were more likely to report lower extremity pain only, possibly because they perform work in a standing position almost all day long, in contrast to other non-manual workers [37].

Our study showed that exposure to a combination of ergonomic risk factors at the workplace had adverse effects, and that providing workers with adequate time to rest reduced these adverse effects. Armstrong et al. [3] presented a conceptual model to explain the complex relationship of WRMSDs with not only ergonomic risk factors such as the amount, duration, and frequency of physical loads but also time for rest and recovery. The present study highlighted the complex and dynamic relationship of musculoskeletal problems with co-exposure to ergonomic risk factors and enough time for recovery. Our results thus have important occupational health implications. To reduce musculoskeletal symptoms among workers, employers should consider implementing interventions or programs that change the conditions of execution of the work, and thus protect employees from exposure to a combination of ergonomic risk factors responsible for musculoskeletal symptoms. Employers should also provide sufficient time for rest or recovery to allow their fatigued muscles to recover.

The major strength of our analysis is that the data were from the KWCS, a survey that is representative of all Korean adult workers that used rigorous quality control procedures [16]. Our study also had some weaknesses. First, because this was a cross-sectional analysis, we cannot infer causal relationships between ergonomic risk factors with musculoskeletal symptoms in this cross-sectional analysis. It is possible that unknown intermediary factors were responsible for the reported associations. Nonetheless, a causal relationship between adverse work conditions and musculoskeletal symptoms seems highly plausible. A second limitation is that we relied on self-reported data instead of objective findings. Thus, our results should be interpreted cautiously.

## 5. Conclusions

Our major findings are that workers who were exposed to a greater number of ergonomic risk factors had increased odds ratios for musculoskeletal symptoms, and that workers who were provided with sufficient time for rest and recovery had reduced odds ratios for musculoskeletal symptoms. Thus, employers should provide comprehensive interventions that change the work, and thus, protect their employees from exposure to multiple ergonomic risk factors responsible for musculoskeletal symptoms. Employers should also provide their employees with sufficient time for rest and recovery of fatigued muscles.

## Figures and Tables

**Table 1 ijerph-17-09456-t001:** Socio-demographic characteristics of study subjects (N = 30,065).

Variable	Classification	No (%)
Gender	Men	14,479 (48.2)
	Women	15,586 (51.8)
Age, years	<40	10,953 (36.4)
	40–49	7744 (25.8)
	50–59	6894 (22.9)
	>60	4474 (14.9)
Education	<High school	3773 (12.6)
	High school	10,493 (34.9)
	>High school	15,776 (52.5)
Monthly income, USD	<1000	3023 (10.1)
	1000–<2000	9214 (30.9)
	2000–<3000	8608 (28.9)
	3000–<4000	5285 (17.7)
	≥4000	3671 (12.3)
Occupational class	Managers, professionals, and clerks	12,065 (40.3)
	Service and sales workers	7912 (26.4)
	Skilled manual workers	5389 (18.0)
	Unskilled manual workers	4606 (15.4)
Weekly working hours	<40	4641 (15.5)
	40–<48	16,002 (53.3)
	48–<60	6312 (21.0)
	≥60	3045 (10.2)

**Table 2 ijerph-17-09456-t002:** The proportion of musculoskeletal pain by demographic, socioeconomic, and work factors (N = 30,065).

Variable	Classification	Back Pain		Upper Extremity Pain		Lower Extremity Pain	
		No (%)	*p*-Value	No (%)	*p*-Value	No (%)	*p*-Value
Gender	Men	1328 (9.2)	<0.001	2883 (19.9)	<0.001	1772 (12.2)	<0.001
	Women	1940 (12.5)		3891 (25.0)		2731 (17.5)	
Age, years	<40	586 (5.4)	<0.001	1448 (13.2)	<0.001	830 (7.6)	0.001
	40–49	728 (9.4)		1641 (21.2)		949 (12.3)	
	50–59	956 (13.9)		2019 (29.3)		1373 (19.9)	
	>60	998 (22.3)		1666 (37.3)		1351 (30.2)	
Education	<High school	1016 (26.9)	<0.001	1701 (45.1)	<0.001	1383 (36.7)	<0.001
	High school	1295 (12.3)		2842 (27.1)		1892 (18.0)	
	>High school	955 (6.1)		2223 (14.1)		1222 (7.7)	
Monthly income, USD	<1000	609 (20.2)	<0.001	945 (31.3)	<0.001	777 (25.7)	<0.001
	1000–<2000	1173 (12.7)		2402 (26.1)		1705 (18.5)	
	2000–<3000	771 (9.0)		1816 (21.1)		1087 (12.6)	
	3000–<4000	405 (7.7)		915 (17.3)		545 (10.3)	
	≥4000	276 (7.5)		628 (17.1)		348 (9.5)	
Vibration	No	2598 (9.9)	<0.001	5321 (20.4)	<0.001	3588 (13.7)	<0.001
	Yes	668 (17.1)		1450 (37.1)		912 (23.4)	
Painful or tiring postures	No	1317 (6.4)	<0.001	3072 (14.9)	<0.001	1961 (9.5)	<0.001
	Yes	1951 (20.8)		3701 (39.5)		2540 (27.1)	
Lifting heavy loads	No	2438 (9.6)	<0.001	5033 (19.8)	<0.001	3309 (13.0)	<0.001
	Yes	830 (18.0)		1741 (37.7)		1193 (25.8)	
Repetitive hand or arm movements,	No	1029 (7.9)	<0.001	1999 (15.4)	<0.001	1414 (10.9)	<0.001
	Yes	2239 (13.1)		4773 (27.9)		3086 (18.1)	
Occupational class	Managers, professionals, and clerks	669 (5.5)	<0.001	1596 (13.2)	<0.001	816 (6.8)	<0.001
	Service and sales workers	794 (10.0)		1722 (21.8)		1286 (16.3)	
	Skilled manual workers	827 (15.4)		1753 (32.5)		1062 (19.7)	
	Unskilled manual workers	974 (21.2)		1699 (36.9)		1334 (29.0)	
Weekly working hours	<40	782 (16.9)	<0.001	1377 (29.7)	<0.001	1091 (23.5)	<0.001
	40–<48	1321 (8.3)		2829 (17.7)		1661 (10.4)	
	48–<60	678 (10.7)		1605 (25.4)		1074 (17.0)	
	≥60	475 (15.6)		945 (31.0)		668 (21.9)	
Ability to rest when desired	Yes	674 (7.6)	<0.001	1464 (16.5)	<0.001	958 (10.8)	<0.001
	No	2593 (12.2)		5306 (25.1)		3541 (16.7)	
Having time less than 11 h prior to next shift	No	2918 (10.4)	<0.001	6182 (22.0)	<0.001	4072 (14.5)	<0.001
	Yes	339 (17.9)		574 (30.2)		415 (21.8)	

**Table 3 ijerph-17-09456-t003:** Multivariate Logistic analysis of the association of musculoskeletal pain with combined ergonomic hazards, demographic factors, socioeconomic factors, and other factors in workers.

		OR (95%CI)		
Variable	Classification	Back Pain	Upper Extremity Pain	Lower Extremity Pain
No. of ergonomic hazards	0	1.0 (reference)	1.0 (reference)	1.0 (reference)
	1	1.498 (1.335–1.680) ***	1.667 (1.536–1.809) ***	1.500 (1.361–1.654) ***
	2	2.918 (2.602–3.273) ***	3.048 (2.796–3.323) ***	2.382 (2.154–2.635) ***
	3	3.252 (2.842–3.722) ***	4.071 (3.669–4.516) ***	3.283 (2.918–3.693) ***
	4	3.391 (2.843–4.045) ***	4.938 (4.303–5.668) ***	3.893 (3.336–4.543) ***
Gender	Women vs. Men	1.563 (1.418–1.724) ***	1.602 (1.487–1.725) ***	1.685 (1.545–1.838) ***
Age, years	<40	1.0 (reference)	1.0 (reference)	1.0 (reference)
	40–49	1.713 (1.520–1.930) ***	1.628 (1.497–1.770) ***	1.509 (1.360–1.676) ***
	50–59	1.980 (1.751–2.238) ***	1.909 (1.748–2.085) ***	1.940 (1.746–2.157) ***
	≥60	2.243 (1.930–2.607) ***	1.872 (1.667–2.102) ***	2.149 (1.885–2.451) ***
Education	<High school	1.0 (reference)	1.0 (reference)	1.0 (reference)
	High school	0.575 (0.509–0.649) ***	0.561 (0.506–0.622) ***	0.546 (0.490–0.609) ***
	>High school	0.513 (0.438–0.601) ***	0.464 (0.410–0.526) ***	0.430 (0.374–0.495) ***
Monthly income, USD	<1000	1.0 (reference)	1.0 (reference)	1.0 (reference)
	1000–<2000	0.747 (0.642–0.869) ***	1.019 (0.899–1.156)	0.998 (0.871–1.143)
	2000–<3000	0.684 (0.574–0.816) ***	1.031 (0.895–1.188)	0.938 (0.802–1.097)
	3000–<4000	0.682 (0.557–0.835) ***	0.950 (0.810–1.114)	0.965 (0.805–1.156)
	≥4000	0.738 (0.593–0.920) **	1.078 (0.908–1.279)	1.016 (0.834–1.238)
Job category	Managers, professionals, and clerks	1.0 (reference)	1.0 (reference)	1.0 (reference)
	Service and sales workers	1.057 (0.925–1.206)	1.029 (0.936–1.131)	1.369 (1.221–1.535) ***
	Skilled manual Work	1.564 (1.356–1.803) ***	1.488 (1.341–1.651) ***	1.537 (1.352–1.747) ***
	Unskilled manual Work	1.395 (1.197–1.627) ***	1.312 (1.168–1.474) ***	1.610 (1.405–1.845) ***
Weekly working hours	<40	1.0 (reference)	1.0 (reference)	1.0 (reference)
	40–<48	1.059 (0.921–1.217)	0.949 (0.850–1.058)	0.838 (0.742–0.947) **
	48–<60	1.057 (0.909–1.229)	1.143 (1.017–1.285) *	1.089 (0.956–1.239)
	≥60	1.357 (1.148–1.604) ***	1.317 (1.153–1.504) ***	1.216 (1.050–1.407) **
Ability to rest when desired		0.647 (0.589–0.711) ***	0.624 (0.583–0.669) ***	0.667 (0.615–0.724) ***
Those without time less than 11 h between consecutive shifts		0.495 (0.430–0.568) ***	0.642 (0.572–0.721) ***	0.566 (0.497–0.664) ***

* *p* < 0.05, ** *p* < 0.01, *** *p* < 0.001.

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
