# Peer review of "Association of Exposure to a Combination of Ergonomic Risk Factors with Musculoskeletal Symptoms in Korean Workers"

_ijerph, 2020, doi:10.3390/ijerph17249456_

Round 1

Reviewer 1 Report

The article is well written and presents an excellent methodological outline.
I think they should use the terms MSD and WRMSD always in the plural: MSDs and WRMSDs.
Although I do not consider myself fluent enough in the English language, it seems to me that the use of the designation "rest" and "adequate rest" in the context of your study is fallacious.
In reality, they refer to a situation of organizational imposition of schedules that is not directly related to rest; the 11-hour difference between shifts can be used by workers to rest as for other day-to-day demands and that does not mean they "rested".
So, I suggest that you use "enough time to rest between consecutive shifts".
on the other hand, it seems to me that you excessively value the results of the decrease in musculoskeletal symptoms in relation to the reduced rest time between shifts (less than 11 hours) because it is only a reduced value of aOR of 0.566 (CI 0.497-0.664 ), even if significant.
On lines 163, 165 and 166 there is "smusculoskeletal .."; what does it mean? or is the "s" an error?
Finally, in the discussion you apparently suggest an intervention that, in essence, allows "adequate rest", that is, an organizational intervention and you never mention the need to intervene equally on the work physical and psychosocial demands. It is unclear the need for intervention on working conditions that determine exposure to different ergonomic risk factors. The work cannot be considered as "immutable" and one of the main objectives of Ergonomics is to change the work, in particular the conditions of execution of the work. Only by changing the conditions in which the work is performed it will be possible reducing exposure to the risk factors responsible for musculoskeletal symptoms.

Author Response

Comment) I think they should use the terms MSD and WRMSD always in the plural: MSDs and WRMSDs.

Response) We changed the terms MSD and WRMSD into MSDs and WRMSDs.

Comment) Although I do not consider myself fluent enough in the English language, it seems to me that the use of the designation "rest" and "adequate rest" in the context of your study is fallacious.
In reality, they refer to a situation of organizational imposition of schedules that is not directly related to rest; the 11-hour difference between shifts can be used by workers to rest as for other day-to-day demands and that does not mean they "rested".
So, I suggest that you use "enough time to rest between consecutive shifts".

Response) We changed the terms "rest" and "adequate rest" into "enough time to rest between consecutive shifts".

Comment) on the other hand, it seems to me that you excessively value the results of the decrease in musculoskeletal symptoms in relation to the reduced rest time between shifts (less than 11 hours) because it is only a reduced value of aOR of 0.566 (CI 0.497-0.664 ), even if significant.

Response) We addressed the significance of aOR of 0.495 in Results section.

In particular, those without time less than 11 h between consecutive shifts were 50.5% less likely to report back pain.

Comment) On lines 163, 165 and 166 there is "smusculoskeletal .."; what does it mean? or is the "s" an error?

Response) We corrected errors.

Comment) Finally, in the discussion you apparently suggest an intervention that, in essence, allows "adequate rest", that is, an organizational intervention and you never mention the need to intervene equally on the work physical and psychosocial demands. It is unclear the need for intervention on working conditions that determine exposure to different ergonomic risk factors. The work cannot be considered as "immutable" and one of the main objectives of Ergonomics is to change the work, in particular the conditions of execution of the work. Only by changing the conditions in which the work is performed it will be possible reducing exposure to the risk factors responsible for musculoskeletal symptoms.

Response) We addressed the significance of changing the work in Discussion section.

To reduce musculoskeletal symptoms among workers, employers should consider implementing interventions or programs that change the conditions of execution of the work, and thus, protect employees from exposure to a combination of ergonomic risk factors responsible for musculoskeletal symptoms.

Thus, employers should provide comprehensive interventions that change the work, and thus, protect their employees from exposure to multiple ergonomic risk factors responsible for musculoskeletal symptoms.

Reviewer 2 Report

  1. In Sec. 2.1. There are 30,060 subjects involved in the study. It will be useful to provide the basic data of the selected subjects, such as age, stature, weight, distribution of the career, etc. (because the data is selected out from the original 50,205)
  2. In Sec. 2.2.2 Ln 97-101. What is the reason or standard to divide the subject’s age into the four-level (<40, 40 to 49, 50 to 59, ≥60)? The same question for the category of weekly working hours (<40, 40 to less than 48, 48 to less than 60, ≥60) needs to classify.
  3. Ln 125. Why sleeted “the 11 hours for rest” as a demarcation?
  4. The two major findings of the manuscript stated by authors were 1) workers exposed to more ergonomic risk factors had increased odds ratios for musculoskeletal symptoms, and 2) provided with sufficient time for rest and recovery had reduced odds ratios for musculoskeletal symptoms of workers. However, from the discussion, as we know that the two findings are consistent with the previous studies (Ln 175-186). What are the new insights/contributions of the study?
  1. Discussion section. The discussion section requires revision, including discussing the findings with the existing MSDs literature and pointing out the contributions to environmental research and public health science?
  2. What is the difference between musculoskeletal symptoms and musculoskeletal disorder? In the introduction section, the MSD was used. Spuriously, the “musculoskeletal symptoms” replaced the MSD in the remained content.
  3. Why the authors asked the subject to report the pain history during “the past 12 months”? Some related MSDs studies were recorded the data during the past six months. Why record the feeling during the past 12 months?
  4. Ln 141-145 Please revise the sentences for increasing the readability.
  5. Table 1 is too large. Try to scaled down or rebuild the table.

Author Response

  1. In Sec. 2.1. There are 30,060 subjects involved in the study. It will be useful to provide the basic data of the selected subjects, such as age, stature, weight, distribution of the career, etc. (because the data is selected out from the original 50,205)

Response) We presented a new Table (Table 1) showing socio-demographic characteristics of study subjects. However, stature, weight, and etc. are not available in KWCS data.

Table 1 shows the socio-demographic characteristics of the study subjects. A total of 48.2% of the study subjects were men and 37.8% of them were at least 50 years old. Most study subjects graduated from high school (34.9%) or college (52.5%). A total of 40.7% of study subjects had monthly incomes less than 2000 USD. Among the study subjects, 40.1% were “managers, professionals, and clerks”, followed by “service and sales workers” (26.3%), “skilled manual workers” (17.9%), and “unskilled manual workers” (15.3%). A total of 53.2% worked moderate working hours, followed by long working hours (21.0%), short working hours (15.4%), and excessively long working hours (10.1%) (Table 1).

  1. In Sec. 2.2.2 Ln 97-101. What is the reason or standard to divide the subject’s age into the four-level (<40, 40 to 49, 50 to 59, ≥60)?

Response) We added the reason to divide the subject’s age into the four-level.

We divided the subject’s age into young (<40), middle (40 to 49), aging (50 to 59), and aged (≥60) considering comparability and age distribution.

The same question for the category of weekly working hours (<40, 40 to less than 48, 48 to less than 60, ≥60) needs to classify.

Response) We added the reason for the categorization.

Average weekly working hour were classified as short working hours (<40 h), moderate working hours (40–47 h), long working hours (48–59 h), or excessively long working hours (≥60 h). Forty hour per week is the standard working hour, and weekly working hours exceeding 48 are defined as long working hours.[17] Excessively long working hours are defined as ≥60 h in Japan and Korea considering the risk of Karoshi.[18-20]

  1. Ln 125. Why selected “the 11 hours for rest” as a demarcation?

Response) We added the reason for the selection.

Concerning the organisation of working time, Directive 2003/88/EC states that Member States shall take the measures necessary to ensure that every worker is entitled to a minimum daily rest period of 11 consecutive hours per 24-hour period.[23]

  1. The two major findings of the manuscript stated by authors were 1) workers exposed to more ergonomic risk factors had increased odds ratios for musculoskeletal symptoms, and 2) provided with sufficient time for rest and recovery had reduced odds ratios for musculoskeletal symptoms of workers. However, from the discussion, as we know that the two findings are consistent with the previous studies (Ln 175-186). What are the new insights/contributions of the study?

Response) We discussed the present findings in details in the light of novel findings, and clarified the new insights/contributions of the study in Discussion section.

The present study showed that many workers reported exposure to a combination of ergonomic risk factors that could lead to musculoskeletal symptoms. In particular, exposure to a greater number of ergonomic risk factors increased the odds ratio of musculoskeletal symptoms, upper extremity pain, in particular. However, few previous epidemiological studies have investigated the relationship between exposure to a combination of ergonomic risk factors and WRMSDs, and some of the results have been contradictory.[10-14] Our findings confirmed that WRMSDs are more likely to be caused by complex ergonomic hazards. We also found that workers who had enough time to rest during working hours or between consecutive shifts were less likely to report all the musculoskeletal symptoms. However, only a few papers reported the relationship between rest break and musculoskeletal pain. Armstrong et al.[3] reported that an MSD was more likely when a muscle was repeatedly fatigued and not given sufficient time for recovery. Wang et al.[24] found a strong association between the ratio of work time to recovery time within a day and neck-shoulder disorders. No previous paper reported the association between enough time to rest between consecutive shifts and musculoskeletal problems. The present study is the first to evaluate the relationship of musculoskeletal problems with enough time to rest during working hours and between consecutive shifts to the best of our knowledge. Many previous studies found that workers with long working hours had a higher incidence of musculoskeletal symptoms.[25-28] However, longer working hours do not necessarily lead to shorter resting time although excessively long work schedules may limit rest and recovery time. The present findings support this issue of long working hours.

Our study showed that exposure to a combination of ergonomic risk factors at the workplace had adverse effects, and that providing workers with adequate time to rest reduced these adverse effects. Armstrong et al.[3] presented a conceptual model to explain the complex relationship of WRMSDs with not only ergonomic risk factors such as the amount, duration, and frequency of physical loads, but also time for rest and recovery. The present study highlighted the complex, and dynamic relationship of musculoskeletal problems with co-exposure to ergonomic risk factors and enough time for recovery time.

  1. Discussion section. The discussion section requires revision, including discussing the findings with the existing MSDs literature and pointing out the contributions to environmental research and public health science?

Response) We discussed our findings with the existing MSDs literature and pointing out the contributions to environmental research and public health science in Discussion section.

The present study showed that many workers reported exposure to a combination of ergonomic risk factors that could lead to musculoskeletal symptoms. In particular, exposure to a greater number of ergonomic risk factors increased the odds ratio of musculoskeletal symptoms, upper extremity pain, in particular. However, few previous epidemiological studies have investigated the relationship between exposure to a combination of ergonomic risk factors and WRMSDs, and some of the results have been contradictory.[10-14] Our findings confirmed that WRMSDs are more likely to be caused by complex ergonomic hazards. We also found that workers who had enough time to rest during working hours or between consecutive shifts were less likely to report all the musculoskeletal symptoms. However, only a few papers reported the relationship between rest break and musculoskeletal pain. Armstrong et al.[3] reported that an MSD was more likely when a muscle was repeatedly fatigued and not given sufficient time for recovery. Wang et al.[24] found a strong association between the ratio of work time to recovery time within a day and neck-shoulder disorders. No previous paper reported the association between enough time to rest between consecutive shifts and musculoskeletal problems. The present study is the first to evaluate the relationship of musculoskeletal problems with enough time to rest during working hours and between consecutive shifts to the best of our knowledge. Many previous studies found that workers with long working hours had a higher incidence of musculoskeletal symptoms.[25-28] However, longer working hours do not necessarily lead to shorter resting time although excessively long work schedules may limit rest and recovery time. The present findings support this issue of long working hours.

Our study showed that exposure to a combination of ergonomic risk factors at the workplace had adverse effects, and that providing workers with adequate time to rest reduced these adverse effects. Armstrong et al.[3] presented a conceptual model to explain the complex relationship of WRMSDs with not only ergonomic risk factors such as the amount, duration, and frequency of physical loads, but also time for rest and recovery. The present study highlighted the complex, and dynamic relationship of musculoskeletal problems with co-exposure to ergonomic risk factors and enough time for recovery time.

  1. What is the difference between musculoskeletal symptoms and musculoskeletal disorder? In the introduction section, the MSD was used. Spuriously, the “musculoskeletal symptoms” replaced the MSD in the remained content.

Response) Some studies in Introduction section defined health outcome as musculoskeletal disorder, however, other studies in Introduction section defined outcome as musculoskeletal symptoms such as back pain.[12-14] The present study used self-reported musculoskeletal symptoms in questionnaire.

  1. Why the authors asked the subject to report the pain history during “the past 12 months”? Some related MSDs studies were recorded the data during the past six months. Why record the feeling during the past 12 months?

Response) The present study made a secondary analysis of data of the fifth Korean Working Conditions Survey (KWCS), which asked the subject to report the pain history during “the past 12 months”.

  1. Ln 141-145 Please revise the sentences for increasing the readability.

Response) We revise the sentences for increasing the readability.

Musculoskeletal symptoms in the back, upper extremity, and lower extremity were common in the following groups such as female workers, aged workers, low educated and low income workers, manual workers, workers exposed to ergonomic risk factors, and workers with the shortest or longest weekly working hours, no ability to rest when desired, and/or time less than 11 h between consecutive shifts.

  1. Table 1 is too large. Try to scaled down or rebuild the table.

Response) We rebuilt the table in a simple format.

Round 2

Reviewer 2 Report

All my concerns are carefully responses. English can consider to polish before publish.